# Improvement of Barrier Properties of Biodegradable Polybutylene Succinate/Graphene Nanoplatelets Nanocomposites Prepared by Melt Process

**DOI:** 10.3390/membranes11020151

**Published:** 2021-02-22

**Authors:** Raphaël Cosquer, Sébastien Pruvost, Fabrice Gouanvé

**Affiliations:** 1UMR CNRS 5223 IMP Ingénierie des Matériaux Polymères, Univ Lyon INSA Lyon, 69621 Villeurbanne, France; raphael.cosquer@insa-lyon.fr (R.C.); sebastien.pruvost@insa-lyon.fr (S.P.); 2UMR CNRS 5223 IMP Ingénierie des Matériaux Polymères, Univ Lyon Université Lyon 1, 69622 Villeurbanne, France

**Keywords:** graphene nanocomposites, polybutylene succinate, water sorption, water and dioxygen permeability, melt blending

## Abstract

Polybutylene Succinate (PBS)/Graphene nanoplatelets (GnP) nanocomposites over a range of GnP from 0 to 1.35 wt.%. were prepared by a melt process. A mixture of individual graphene nanosheets and aggregates was obtained by the addition of GnP in the PBS matrix. The presence of these fillers did not significantly modify the morphology, crystalline microstructure of the matrix or its thermal stability. However, a slight reinforcement effect of PBS was reported in the presence of GnP. The water sorption isotherm modelling with Guggenheim, Andersen and De Boer (GAB) equation and Zimm-Lundberg theory allowed a phenomenological analysis at the molecular scale. The presence of GnP did not modify the water sorption capacity of the PBS matrix. From a kinetic point of view, a decrease of the diffusion coefficient with the increasing GnP content was obtained and was attributed to a tortuosity effect. The influence of water activity was discussed over a range of 0.5 to 1 and 0 to 0.9 for water and dioxygen permeability. Improvement of the barrier properties by 38% and 35% for water and dioxygen permeability respectively were obtained.

## 1. Introduction

Biodegradable polymers-based nanocomposites are an interesting alternative to the usual non-biodegradable polymer regarding global environmental problems caused by the amount of plastic waste [1]. A large panel of biodegradable polyesters, such as poly (L-lactic acid) (PLA), poly(ε-caproloctone) (PCL), polybutylene succinate (PBS), have been studied in terms of thermal, mechanical and transport properties whether in their neat or their nanocomposites versions [2,3,4,5]. Due to its bio-based production ranging from 0% to 100%, polybutylene succinate (PBS) has been largely studied over the past decades. It is an aliphatic, semi-crystalline, thermoplastic polyester synthesized by polycondensation of succinic acid and 1,4-butanediol, which can both be bio-sourced. PBS has interesting properties, similar to polyethylene (PE) [1], including a solvent-free melt processability, chemical and thermal resistance, good mechanical (good tensile strength and impact strength, moderate rigidity, and hardness), relatively good barrier properties (water and dioxygen) and biodegradability thanks to hydrolysis in enzymatic and environmental conditions (bacteria, fungi, etc.) [6]. In its neat version, PBS cannot meet all requirements, mainly due to the limited barrier properties to water permeation. Improvement of barrier properties is thus required. A promising solution to overcome these limitations is found in the addition of high aspect ratio nanofillers in the polymer. The addition of lamellar filler allows a significant reduction of the permeability of the polymer by inducing a tortuous pathway in the amorphous phase, hence increasing the barrier properties [7,8,9]. However, many studies demonstrate that a good exfoliation of the lamellar filler and good interfaces between the fillers and the polymer matrix are key parameters for significantly improving barrier properties [10,11]. In most studies, lamellar nanofillers can originate from organoclay (native montmorillonite, organo-modified montmorillonite) [3,12,13]. Charlon et al. showed a water permeability improvement of about 35% for 5 wt.% of organo-modified montmorillonite in PBS compared to the neat matrix [8]. Usually, inorganic filler succeeds in improving the barrier properties of polyester based nanocomposites but a relatively high weight percentage of inorganic material is required.

Lamellar nanofillers with a high aspect ratio can also originate from carbon-based derivatives (graphite, graphene, graphene oxide, carbon nanotubes, etc.) and also gives good improvement of thermal, electrical, mechanical properties and barrier properties [14,15] when added to a polymer matrix [16,17,18]. Improvement of properties of a large range of polymer, such as PLA (45% with 1.37%vol of graphene oxide nanosheet, 67% with 0.4 wt.% of graphene oxide and 68% with 0.4% of graphene nanoplatelets on dioxygen permeability) [2,19], polyimide (80% with 0.01 wt.% of graphene oxide on water permeability) [20], poly(vinyl alcohol) (98% and 68% with 0.72%vol of graphene oxide nanosheet on dioxygen and water permeability, respectively) [21], by the addition of well dispersed graphene or graphene oxide has been studied. Carbon based fillers with a high aspect ratio seems to be good candidates for the improvement of barrier properties.

Three main processes are used in order to elaborate nanocomposites. The first two, in situ and solvent processes are well known to obtain good dispersion. However, their use is limited in an industrial process due to the use of harmful solvent and/or chemical step [16,22]. The third main way is the melt process. This way has the advantage of being a solvent free process and allows the possibility of easy industrialized production.

PBS/carbon based nanofiller nanocomposites have been studied over the past few years in terms of morphology, crystallization, degradation, mechanical and thermal properties, and so forth. [16,17,22,23]. It is worth noticing that, in every case, the best dispersion, hence the reduction of permeability was achieved with a solvent process rather than a melt process. To date, PBS/Graphene or PBS/Graphene nanoplatelets (GnP) nanocomposites have only been discussed a few times in the literature [24,25,26] and most of the time with a solvent process. In these studies, various properties cited previously have been reviewed but the improvement of the transport properties of PBS by the addition of graphene or GnP has not been discussed in the literature.

The novelty of this paper deals with the understanding of the water sorption mechanism of PBS polymer from a thermodynamic point of view using phenomenological models (GAB model and the Zimm-Lundberg theory). We also tried to correlate the obtained results of structural characterization and the obtained results of transport properties. Moreover, dioxygen and water permeation analyses were also performed and these molecules were used as a probe to characterize the architecture of the different films in order to evaluate the influence of the presence of the graphene fillers on the transport properties of PBS nanocomposites films. It is an original approach to establish the relationships between the nanostructure, morphology and transport properties of biodegradable based materials.

## 2. Materials and Methods

### 2.1. Materials

A commercial, 50% bio-sourced, (reference *PBE 003 BB*) polybutylene succinate (PBS) was supplied by Natureplast© (Ifs, France). The chemical structure of the PBS repeating unit is displayed in Figure 1.

Graphene nanoplatelets (GnP) with an average of 15 µm length and a 11–15 nm thickness were provided by SkySpring Nanomaterials© (Houston, TX, USA) under the reference 0544DX Graphene nanopowder.

### 2.2. Film Processing

Prior to the melt process, PBS pellets were dried overnight at 80 °C in a vacuum oven to remove traces of sorbed water molecules in order to avoid potential hydrolytic degradation of the PBS polymer chains. In a first step, PBS pellets were added to an internal mixer, an HAAKE Minilab micro-compounder from Thermo Scientific© (Waltham, MA, USA) for 2 min at 170 °C. The rotor speed was adjusted at 50 rpm. Then, GnP powder was added and blended for 3 min more in the same conditions. At the end of this phase, a nanocomposite masterbatch with a theoretical GnP amount of 5 wt.%. was obtained. In parallel, the same process was applied to neat PBS pellets for a total of 5 min to ensure that the PBS strands used for the following dilution of the masterbatch had the same thermal history. After that, the masterbatch and neat PBS strands were cooled in liquid nitrogen and then grained using a Fritsch© (Idar-Oberstein, Germany) Power cutting Mill Pulverisette 25. The obtained pellets were weighted and hand mixed, depending on the targeted weight percentage, and were extruded using a double screw Micro compounder Xplore© (Sittard, The Netherlands) MC 15 HT at 170 °C for 5 min and then injected into 2 mm thick disks using an injection molder Xplore IM 12 at 150 °C and two steps of pressure: 8 bar for 3 s and 5 bar during 10 s. Disks were finally hot pressed to obtain films of about 100 µm thickness with a Polystat 200 T of Servitec© (Wustermark, Germany) at 150 °C and 150 bar. Films were cooled from 150 °C to ambient temperature (25 °C) without control allowing a slow cooling rate. The same process conditions were used to perform neat PBS films. Theoretical compositions and the sample reference of the prepared films are summarized in Table 1. The real GnP amount in the nanocomposite films will be discussed later.

### 2.3. Steric Exclusion Chromatography

The average molecular mass of the neat PBS matrix and nanocomposites was determined by Steric Exclusion Chromatography (SEC). Solutions were prepared by dissolving each sample in chloroform with a concentration of about 1 mg mL^−1^ and then filtered with a 0.45 µm polytetrafluoroethylene filter. Analyses were performed on a CTO 20A coupled with a LC-20AD pump from Shimadzu© (Kyoto, Japan). Detectors used were a Reflective Index Detector RID-10A from Shimadzu© (Kyoto, Japan) and a Viscostar II^®^ coupled with a miniDAWN™ TREOS^®^ from WYATT technology© (Santa Barbara, CA, USA). The d*n*/d*C* of PBS was taken as 0.05 mLg^−1^ [27].

### 2.4. Differential Scanning Calorimetry

Differential Scanning Calorimetry (DSC) experiments were carried out with a Q200 apparatus from TA Instrument© (New Castle, DE, USA) under a nitrogen atmosphere following a three steps process. Samples were heated from − 70 °C to 150 °C, then cooled from 150 °C to − 70 °C and finally heated again from − 70 °C to 150 °C. For all three steps, the rate was 10 °C min^−1^. The crystallinity index ($X_{C}$) was calculated using the following equation:(1)$$X_{C}=\frac{\Delta H_{m}}{\Delta H_{m}^{0}\cdot \left(1-\varphi \right)}$$
where *φ* is the mass fraction of GnP added in the nanocomposite, $\Delta H_{m}$ is the melting enthalpy of the sample, $\Delta H_{m}^{0}$ is the extrapolated value of the enthalpy corresponding to the melting of theoretical 100% crystalline pure PBS, which is taken as 200 J.g^−1^ [22,26,28]. Each value measured was displayed as the average value for at least three samples.

### 2.5. Wide-Angle X-Ray Scattering in Reflection Mode

Wide-angle X-ray scattering (WAXS) in reflection mode analyses were carried out at room temperature using a Cu tube (*λ* = 1.5406 Å, 40 kV, 40 mA and a nickel filter in order to remove the Kβ line) and a Bruker© (Billerica, MA, USA) D8 Advance diffractometer with a Bragg-Brentano configuration. The diffraction patterns were obtained in the *2θ* range between 5° and 50 by steps of 0.02°. Films were deposited on neutral monosubstrates with a thin transfer adhesive with a low scattering response. GnP (powder) was also deposited on neutral monosubstrates with a perfect flat surface. Peaks deconvolution was realized using Fityk software with a Gaussian method.

### 2.6. Thermo-Gravimetric Analysis

Thermo-gravimetric analysis (TGA) analyses were carried out with a TGA 1 STAR System coupled with a Gas Controller GC 200 STAR System from Mettler Toledo© (Columbus, OH, USA). Samples were heated from room temperature to 800 °C with a heating rate of 20 °C min^−1^ under air atmosphere.

### 2.7. Transmission Electron Microscopy (TEM)

Dispersion of GnP was analyzed by transmission electron microscopy (TEM) with a Jeol© (Yokyo, Japan) JEM-1400 Flash equipped with Gatan© (Pleasanton, CA, USA) RIO 16 Mpx camera operated at an accelerating voltage of 120 kV and were performed at the “Centre Technologique des Microstructures” of the University Claude Bernard Lyon1. All samples were prepared from films of about 100 µm thick by cryo-ultramicrotomy at temperature of −90 °C, using a UC7 LEICA© (Wetzlar, Germany) ultramicrotome, to obtain slices with a thickness of about 80 nm, and were deposited on copper grids (CF200-Cu from EMS) covered by a thin layer of carbon.

### 2.8. Tensile Test

Tensile tests were performed using a uniaxial tensile test bench Autograph AGS-X with a 10 kN captor from Shimadzu© (Kyoto, Japan). Tests were carried out at 25 °C on films with H3 type tensile specimens. The crosshead speed was adjusted to 10 mm min^−1^. Values of tensile modulus (*E*), yield stress (*σ_y_*), yield strain (*ε_y_*), breaking stress (*σ_b_*), and strain at break (*ε_b_*) were determined from the stress-strain curves. The reported values of the mechanical characteristics were the arithmetic mean of at least 10 different specimens.

### 2.9. Dynamic Mechanical Analysis (DMA)

Dynamic mechanical analysis (DMA) is a widely used technique for determination of the viscoelastic behavior of polymeric materials. Three main sources of information can be extracted from the DMA test. The storage modulus defines the elastic portion of the stored energy (*G*′), the loss modulus represents the dissipated energy of polymer (*G*^″^) and the tan *δ* gives information on the temperature related transition/relaxation in the polymer (*G*^″^/*G*′). Samples with a beam form were precisely measured (about 17.5 mm × 4.0 mm × 2.0 mm) prior to the experiment. All samples were tested under air atmosphere at a 20 µm displacement amplitude over −90 °C and 90 °C with a heating rate of 2 °C min^−1^ at 10 Hz on a Q800 from TA Instrument© (New Castle, DE, USA).

### 2.10. Dynamic Vapor Sorption (DVS)

Water sorption isotherms of the different films were determined at 25 °C by using the dynamic vapor sorption analyzer, Dynamic Vapor Sorption (DVS) (London, UK) Advantage. Each sample was predried in the DVS Advantage by exposure to dry nitrogen until the equilibrated dry mass was obtained (*m_0_*). A partial pressure of vapor (*p*) was then established within the apparatus by mixing controlled amounts of dry and saturated nitrogen and the mass of the sample (*m_t_*) was followed as a function of time. The mass of the sample at equilibrium (*m_eq_*) was considered to be reached when changes in mass with time (*dm/dt*) were lower than 2.10^−4^ mg min^−1^ for at least 5 min. Then, vapor pressure was increased in suitable activity up to 0.9 by step of 0.1. The value of the mass gain at equilibrium (*M*) defined as $\left(m_{eq}-m_{0}\right)/m_{0}$ for each water activity (*a_w_*) allowed to plot the water sorption isotherm for each sample.

The sorption rate was also estimated at each water activity firstly by fitting the sorption data using an empirical equation:(2)$$\frac{m_{water\;t}\;}{m_{water\;eq}}=kt^{n}$$
where *k* is a constant and *n* is a value indicating the type of diffusion mechanism. Three cases can be considered for the *n* value. The first one *n* = 0.5, corresponding to a Fickian transport, the rate of diffusion is much lower than the rate of relaxation. In the second one, the diffusion is very fast, contrary to the rate of relaxation, and *n* = 1. The third case corresponds to an anomalous diffusion with *n* values lying between 0.5 and 1 [29].

After checking the *n* values, Fick’s diffusion law was applied when *n* = 0.5. Considering the film thickness (*L*), the water diffusion coefficient (*D*) was calculated for the short time (*t*) according to the following equation:(3)$$\frac{m_{water\;t}\;}{m_{water\;eq}}=\frac{4}{L}{\left(\frac{D.t}{\pi }\right)}^{0.5}$$

$m_{water\;t}$ is the mass of water sorbed as a function of the time and $m_{water\;eq}$ is the mass of water sorbed at equilibrium for a given water activity. The precision of the values of the water mass gain at equilibrium and values of the diffusion coefficient was estimated to be better than 5%.

### 2.11. Water Permeation

Water permeability measurements were performed on a Mocon© (Minneapolis, MI, USA) Permatran W3/33 equipped with an infrared sensor. The detector was calibrated by using polyethylene terephthalate films. The test cell was composed of two chambers separated by a film. Prior to testing, films were conditioned in nitrogen atmosphere in the unit for at least 12 h to remove traces of atmospheric water vapor. Water molecules in a vapor or liquid state were introduced to the upstream compartment of the test cell. Water transferred through the film was conducted by the carrier N_2_ gas to the infrared sensor.

The water permeability coefficient ($P_{H_{2}O}$) was calculated considering the following equation:(4)$$P_{H_{2}O}=\frac{J_{st_{H_{2}O}}.L}{\Delta p}$$
where L is the thickness of the film, $J_{st_{H_{2}O}}\;$is the water stationary flux and Δp the difference of pressure between the upstream and the downstream compartments of the permeation cell. $P_{H_{2}O}$ values were expressed in barrer (1 Barrer = 10^10^
${\mathrm{cm}}_{\mathrm{STP}}^{3}$ cm cm^2^ s^−1^ cmHg^−1^) = 3.36 × 10^−16^ mol m^−1^ m^−2^ s ^−1^ Pa^−1^) and the precision on the obtained values was estimated to be better than 5%. $P_{H_{2}O}$ were determined at different water activities, *a_w_* = 0.5, 0.7, 0.8 and 1.

### 2.12. Dioxygen Permeation

Dioxygen permeability measurements were performed on a Mocon© (Minneapolis, MI, USA) Oxtran 2/21 equipped with a colorimetric sensor. The test cell was composed of two chambers separated by the film. Nitrogen containing 2% of hydrogen (N_2_/H_2_) was used as the carrier gas and pure dioxygen was used as the test gas. The water activity of the two gases was controlled by a humidifier. Prior to testing, films were conditioned in N_2_/H_2_ atmosphere in the unit for at least 12 h on the one hand to remove traces of atmospheric dioxygen and on the other hand to be at the water uptake equilibrium condition of the film. Then, dioxygen was introduced in the upstream compartment of the test cell. O_2_ molecules transferred through the film were conducted by the carrier N_2_/H_2_ gas to the coulorimetric sensor. A steady-state line was obtained after a transitory state.

The dioxygen permeability coefficient ($P_{O_{2}}$) was calculated considering the following equation:(5)$$P_{O_{2}}=\frac{J_{st_{O_{2}}}.L}{\Delta p}$$
where L is the thickness of the film, $J_{st_{O_{2}}}\;$ is the dioxygen stationary flux and Δp the difference of pressure between the upstream and the downstream compartments of the permeation cell. $P_{O_{2}}$ values were expressed in barrer (1 Barrer = 10^10^
${\mathrm{cm}}_{\mathrm{STP}}^{3}$ cm cm^2^ s^−1^ cmHg^−1^) = 3.36 × 10^−16^ mol m^−1^ m^−2^ s ^−1^ Pa^−1^) and the precision on the obtained values was estimated to be better than 5%. $P_{O_{2}}$ were dertermined at controlled temperature, T = 25 °C and different water activity, *a_w_* = 0.5, 0.7, 0.8 and 1.

## 3. Analysis

Sorption models are useful to predict the type of interactions involved throughout the sorption process. GAB model (Guggenheim, Anderson, de Boer) is usually used to fit BET type II and BET type III isotherm curves [30,31,32]. The GAB equation describing isotherms is based on the assumption of localized physical adsorption in multilayers with no lateral interactions. According to this theory, the first molecules are sorbed very strongly in the monolayer. Once this monolayer is reached, the molecules will subsequently be sorbed with weaker interactions with the sorbent surface and the range in energy levels is between those of the monolayer molecules and the bulk liquid. The equation of GAB model introduces three physical parameters (*Mm*, *C_G_* and *K*) and is defined as:(6)$$M=M_{m}\frac{C_{G}\cdot K\cdot a_{w}}{\left(1-K\cdot a_{w}\right)\left(1+\left(c_{G}-1\right)\cdot K\cdot a_{w}\right)}$$
where *M_m_* is the monolayer value and describes the availability of active sites for permeant molecules by the polymer. *C_G_* is the Guggenheim constant and defines the strength of binding of water molecule to the primary binding sites. *K* is a correction factor, since it corrects the properties of the multilayer molecules relative to the bulk liquid [33].

To evaluate the accuracy of the GAB model to describe the water sorption isotherms of the studied films, the mean relative percentage of deviation modulus (MRD) is usually determined and defined as:(7)$$MRD\left(\%\right)=\frac{100}{N}\overset{N}{\underset{i=1}{\sum }}\frac{\left|m_{i}-m_{pi}\right|}{m_{i}}$$
where *m_i_* is the experimental value, *m_pi_* is the predicted value, and *N* is the number of experimental data.

The mean relative percentage deviation modulus (*MRD*) is widely adopted through the literature, and a modulus value below 10% indicates a good fit [34]. GAB parameters were determined by fitting according to OriginLab software (OriginLab Corporation, Northampton, MA, USA). Therefore, each isotherm was described analytically by GAB’s equation.

The convex shape of the curve at high activities is generally explained by a formation of water clusters. From the shape of the isotherm, Zimm and Lundberg have developed a method, based on statistical mechanisms which analyses the cluster phenomenon and allows an interpretation of the solution thermodynamic behavior in geometric isotherms [35]. Their method gives an interpretation of the solution thermodynamic behavior in geometric isotherms. Neglecting the isothermal compressibility of polymer-permeant solution makes the free energy function of the system essentially dependent upon the first derivative of the activity with respect to the permeant volume fraction. The elaborated relation appears as follow:(8)$$\frac{G_{S}}{V_{w}}=-\left(1-\phi_{w}\right)\left[\left(\frac{\delta \left(\frac{a_{w}}{\phi_{w}}\right)}{\delta a_{w}}\right)\right]-1$$
where G_s_ is the cluster integral, V_w_, and $\phi_{w}$ are respectively the partial molecular volume and the volume fraction. A *G_S_/V_w_* value equal to −1 indicates that water dissolves into polymer matrix randomly, instead higher values, *G_S_/V_w_* > 1 mean that the concentration of water in the neighborhood of a given water molecule is greater than the average concentration of water molecules in the polymer. The quantity *G_S_Φ_w_/V_w_* is the mean number of molecules in excess of the mean concentration of water in the neighborhood of a given molecules. Thus, the mean cluster size (MCS) can be evaluated by the following equation:(9)$$MCS=1+\left(\frac{\phi_{w}.G_{S}}{V_{w}}\right)$$

*MCS* values can be calculated from GAB parameters considering the following equation:(10)$$MCS=\frac{{\left(\rho_{w}/\rho_{p}\right)}^{2}}{M^{2}{\left(1+\frac{\rho_{w}/\rho_{p}}{M}\right)}^{2}}\times \left[1-\frac{M}{M_{m}\times C_{G}}\left(-2.K.a_{w}(C_{G}-1\right)-2+C_{G}\right]$$
where *ρ_w_* the water density and *ρ_p_* the density of the polymer (PBS = 1.18 g/cm^3^ [26]), *M_m_*, *C_G_*, *K* are the three GAB parameters as explained above and *M* the mass gain at equilibrium.

## 4. Results and Discussion

Film preparation requires several steps, so the impact of the presence of GnP on the PBS polymer backbone during the melt process were evaluated by SEC measurements. The number average molecular weight $\bar{M_{n}}$ and dispersity *Đ* were determined for neat matrix and different nanocomposites. The obtained values are listed in Table 2.

Values of $\bar{M_{n}}$, and *Đ* for neat PBS, PBS/GnP0.1 and PBS/GnP0.5 were close, showing no effect of the presence GnP on chains length of PBS. For higher content (1 and 2 wt.%.), a slight decrease of $\bar{M_{n}}$, and *Đ* was observed, leading to chain scissions in the PBS backbone. Since it is generally admitted that the degradation of the polyester chains via hydrolysis reactions take place preferentially in the amorphous region of the matrix [36], these chain scissions might lead to a reduction of the glass transition temperature (*Tg*). However, the value of $\bar{M_{n}}$ was not drastically reduced because the ratio $\frac{\bar{M_{n}}\left(PBS/GnP2\right)}{\bar{M_{n}}\left(PBS\right)}$ was only 0.8.

### 4.1. Morphology

TEM analysis was used to investigate the dispersion state of GnP nanoplatelets in the PBS composites. The micrographs are shown in Figure 2 and micrographs taken with a lower magnification are presented in Appendix A (Appendix A). Similar dispersion level was found regardless the amount of GnP. Incorporation of GnP into the PBS matrix led to nanocomposites in which the dispersion of the nanofillers can be considered good enough knowing the initial particle/aggregate size (average of 25 µm, measured by Scanning Electron Microscopy) (Appendix A). Some aggregates of hundreds of nanometers or more, consisting in graphene platelets as well as individual graphene nanosheets (indicated with arrows in Figure 2) were observed within the PBS matrix. Moreover, the coexistence of small dispersed graphene layers and tactoids consisting of various layers could be a sign of good interactions of these GnP with the polymer. Similar results were reported by Fukushima et al. for the incorporation of expanded graphite into a PLA polymer matrix [37].

Wide angle X-ray scattering analysis (WAXS) is a powerful tool for examining the structure of polymer nanocomposites. In our case, WAXS was firstly used to check the crystalline nature of the PBS matrix and secondly to determine interlayers distance of GnP. WAXS patterns of the neat matrix and different nanocomposites films are presented in Figure 3A). For clarity, the patterns shown here were shifted vertically and intensity of GnP pattern was divided by 10. The WAXS patterns of the neat PBS and its nanocomposites samples were quite similar. This indicated that the crystalline matrix had the same crystal structure, an α-form of the PBS crystal. The α-form crystal displayed four main diffractions peaks at 19.6°, 21.9°, 22.7°and 29.0° corresponding to respectively (020), (021), (110) and (111) planes respectively [38]. The presence of GnP did not drastically modify the crystalline structure of the PBS matrix. The diffractogram of GnP showed a very intense and narrow peak at 26.6 ° referring to X-ray reflection on the (002) planes of well-ordered graphite. The intensity of this peak increased as the amount of GnP in nanocomposites increased. This plane is generally used to evaluate the exfoliation quality of carbon-based fillers [39]. Corresponding to the perpendicular plane to filler, Bragg’s Law can be used to calculate the *d_spacing_* between two sheets of GnP. Higher value of *d_spacing_* was obtained for lower value of 2*θ*. In our case, from Bragg’s Law a value of *d_(002)_* = 0.33 nm was obtained and corresponded to a graphite-type [40,41]. This seemed to be consistent with the visual aggregate seen on microscopy.

A deconvolution procedure was applied on the XRD patterns according to the position of the different peaks defined previously, using the opensource software Fityk. The diffraction pattern can be decomposed into a broad amorphous halo and peaks from α crystalline phase [42]. The result of the deconvolution curves of the neat matrix as example is shown in Figure 3B and allows a quantification of the crystallinity index (*X_c-WAXS_*). The obtain value of *X_c-WAXS_* for neat PBS matrix was equal to 49%. This value was in agreement to the value found by Phua et al. who also found a value of *X_c_* of 49% by WAXS measurement [43]. A slight decrease of *X_c-WAXS_* from 49 to 47% was obtained with addition of GnP whatever the GnP loading. The slight decrease was attributed to the uncertainty of the measurement of the area of the ($12\bar{1}$) plane since it appears at almost the same 2*θ* as the (002) plane of GnP. Overall, the crystallinity index of PBS measured by WAXS was considered to be unchanged after the addition of GnP. In order to confirm the results obtained by WAXS analysis and have more information relating to the crystalline morphology of the films and the chains mobility of the amorphous phase, these specimens were also analyzed by differential scanning calorimetry.

Chains mobility of the amorphous phase and crystalline structure of PBS matrix are discussed from DSC curves (Figure 4). The values of glass transition temperature (*Tg*) of neat matrix and different nanocomposites are listed in Table 3. The obtained value for PBS was found as −35 °C. Bhatia et al. found a similar value of *Tg* (around −34 °C) [44]. After the incorporation of GnP, a small decrease of *Tg* of the PBS matrix was observed and may be due to the reduction of molar mass as seen from SEC analysis. However, by considering uncertainty on each value, change in *Tg* values were not significant enough to allow strong modification of polymer chains mobility in the amorphous phase. Similar phenomenon has been found by Goncalves et al. on PLA/GnP nanocomposites [11].

Thermograms of the first heating run of PBS neat matrix and nanocomposites show three endothermic melting peaks, labeled as I, II, and III from low to high temperature. The first melting peak (*Tm_I_*) was measured as 35 °C and was not changed after addition of GnP. Makhatha et al. showed that the first endotherm (*Tm_I_*) was not observed when crystallization was performed under non-isothermal condition [45]. In this case, the small endothermwas probably due to the thermal history of the nanocomposites and could be explained by an annealing process occurring at room temperature during the storage of the films since the molecular mobility is high at room temperature (*Tg* = −35 °C). Indeed, this melting peak disappeared for the second heating run (Appendix A). The double endothermic peaks phenomenon (II and III) have been largely discussed in the literature. Several models were proposed to explain the multiple melting behavior of thermoplastic semi-crystalline polymers, of which the most important one is the presence of melting, recrystallization, and remelting phenomena [45,46,47]. According to this model, the first step corresponded to the melting and recrystallization of crystallites with lower thermal stability, followed by the melting of the crystallites with higher thermal stability formed through the recrystallization of the melting of the crystallites of the lower melting endotherms and those already present in the polymer. After the addition of GnP, no modification of *Tm_III_* was observed whereas *Tm_II_* was slightly decreased. Moreover, the distinction between the melting peaks II and III was more apparent as the amount of GnP increased. This phenomenon of peak separation was observed by Makhatha et al. with an increase of the cooling rate [45]. The estimated crystallinity index, measured with melting peaks II and III, of the neat PBS was 38% in the same range order that the values reported by Wang et al. and Pallathadka et al. [22,26]. No modification of *X_c-DSC_* was observed when GnP were added. Similar phenomenon has been found by Goncalves et al. on Polylactic acid (PLA)/GnP nanocomposites [11] showing no significant decrease of *X_c_* in nanocomposites. It might be concluded GnP fillers had consequently no effect on the crystallization of the PBS matrix. Despite the difference in the absolute values obtained by WAXS and DSC methods, both *X_C-WAXS_* and *X_C-DSC_* exhibited the same tendency. As the crystalline lamellae are considered to be impermeable to small molecules, an improvement of barrier properties would not be input to a crystalline change of the PBS matrix in presence of the GnP.

The temperature of the maximum of the peak of crystallization (*T_c_*) of neat PBS was measured as 97 °C. The value of *T_c_* was consistent with those reported in the literature [6,28,47]. Despite the fact that graphitic materials are known to act as nucleating agent when added in PBS [17,48,49], in our case, an increase of GnP loading in nanocomposite led to decrease the temperature of crystallization. *T_onset_*, corresponding to the onset temperature of crystallization and *T_c_* were slightly decreased from 97 °C to 92 °C and 104 °C to 99 °C respectively (Table 3) for the PBS/GnP2 which can be explained by an anti-nucleating effect of the nanofillers. Similar results have been described by Gomari et al. [50] for Poly (ethylene oxide) (PEO)/GnP nanocomposites where an anti-nucleating effect of GnP was partially described and attributed to the ability of GnP to hinder the crystallization. The degree of supercooling ($\Delta T=T_{onset}-T_{c}$) implying the crystallization rate [50] remained unchanged with increasing GnP amount. The full width at the half height maximum of the crystallization peak (FWHM) is considered as an indication of the spherulites size distribution, so the smaller FWHM values demonstrate narrower size distribution [50,51]. A decrease in the FWHM value was obtained when GnP amount increased, indicating a decrease of the size distribution of crystallites. Bhattacharyya et al. showed on Polypropylene/Single Wall Carbon NanoTube (SNWT) nanocomposites, that this effect could be explained, at least partially, by an evenly distribution of heat on the polymer due to the higher thermal conductivity of the carbon nanotubes over the neat polymer [51]. This phenomenon has been noticed in various studies [51,52] with carbon-based filler. This result seemed to be in agreement with the decrease of *Tm_II_* seen on first and second heating step (Figure 4 and Appendix A) that led to a visual separation between both endotherms.

### 4.2. Thermal and Mechanical Properties

Thermogravimetric mass loss curves of the GnP in a powder form are shown in Figure 5. GnP degradation exhibited two mass losses. The first mass loss (around 4%) at 450 °C was attributed to the presence of small quantities of remaining compounds used during the GnP formation. The second mass loss at 650 °C was attributed to the oxidation followed by the degradation of the GnP. This mass loss is commonly seen on graphite-based materials on a dry state under oxidative atmosphere [53].

The typical mass loss curves under air atmosphere for neat PBS matrix and its composites are presented in Figure 5. Two steps were observed on the mass loss curve relative. The first one at about 360 °C and the second one at about 450 °C. Makhatha et al. also showed similar results on PBS [45]. They attributed the first loss to high molecular weight chains decomposition into smaller chain fragments via an initial scissoring of the chain end, followed by the second one, attributed to the following degradation by thermal-oxidation into volatile small molecular products in the presence of dioxygen, of the previous degraded chains.

The amount of GnP in each nanocomposite was deduced from the mass residues measured at 560 °C on the nanocomposites and the neat matrix. This value of 560 °C was chosen because at this temperature both mass loss of PBS was already observed. It also should be noted that the mass loss of GnP at 560 °C was considered for the calculation of GnP loading. The experimental values of GnP loading in mass and volume are listed in Table 4. The value of GnP density, *ρ*_GnP_, was taken as 2.22 g cm^−3^ which is typical for graphitic materials [54]. Lower values than expected ones were obtained because of difficulties faced in the masterbatch preparation due to the powdery form of GnP.

The thermal stability of samples was evaluated by the determination of the temperature of 5% mass loss ($T_{d5\%}$), 50% mass loss ($T_{d50\%}$), and 90% mass loss ($T_{d90\%}$). The obtained values are listed in Table 4. $T_{d5\%,}$
$T_{d50\%}$
$T_{d90\%}$ values of nanocomposites were close compare to those obtained for the neat matrix. A small increase of $T_{d90\%}$ values should be noted, reflecting a slightly lower degradation rate at the end of the chain degradation phenomenon in presence of GnP.

DMA allows measurements of the response of a given material to an oscillatory deformation as a function of temperature. Figure 6 shows the evolution as a function of the temperature of *G*^′^, *G*^″^ and tan *δ*, for neat PBS and corresponding nanocomposites. At low temperature, PBS and its associated nanocomposites were in a glassy state. At the end of the glassy state plateau, a steep decrease on *G*^′^ was observed revealing the apparent glass transition which was in agreement with those found by Makhatha et al. and Yue et al. [45,55]. The steep decrease was followed by a rubbery state described by a slow linear decrease of *G*^′^. A shoulder around 40 °C was also observed and can be explained by the melting of crystallites corresponding to the *Tm_I_* peak underlined from DSC analysis. This shoulder disappeared for the second heating run (Appendix A in Appendix A). The incorporation of GnP did not affect significantly the glass transition temperature. In the same way, the obtained values of *G*^″^ for the nanocomposites regardless the GnP amount were close to those obtained for the PBS matrix, indicating similar internal frictions. The presence of GnP did not lead to significant shift and broadening of the tan *δ* for all nanocomposites compared to that of neat PBS. A maximum of tan *δ* was observed around −16 °C, similar to that reported by Yue et al. [55]. This observed behavior was in agreement to those observed from DSC analyses.

Typical tensile stress-strain curves are presented on Appendix A and Young modulus, stress and strain at break values are summarized in Table 5.

Usually, addition of nanofiller to a polymer matrix leads to an increase of stiffness accompanied with embrittlement. In our case, introduction of nanofillers, irrespective of the amount (except for PBS/GnP0.5) led to a small enhancement of stiffness (13%) and no change of stress at break. However, a small decrease of elongation at break was obtained as the amount of GnP increased. From this result, it might be concluded that the presence of GnP led to a slight reinforcing effect which can be probably attributed to a rather good dispersion of nanofillers in polymer matrix and also to the rather good interfacial adhesion between nanofillers and the polymer matrix.

### 4.3. Water Sorption

Sorption isotherm curves at 25 °C were obtained by plotting the mass gain at equilibrium (*M*) as a function of the water activity (*a_w_*), (Figure 7A). All isotherm curves displayed a BET III shape according to the classification of Brunauer, Emmett, and Teller (BET) [56]. It consists in a linear evolution of water uptake at low water activity (*a_w_*
≤ 0.4) followed by a convex part at high activity (*a_w_* > 0.5). The increase of water uptake at high activity is usually explained by the formation of water clusters [57].

The obtained values of mass gain for neat PBS matrix were in the same order to those obtained by Charlon et al. [3], who found for *a_w_* = 0.9 a mass gain at equilibrium of 0.9%, compared to our obtained value (1.6%). The slight difference can be related to a difference of crystallinity index due to a different thermal history of the films. The water sorption capacity of PBS was in the same range order compared to different polyester polymers such as Polylactic acid (PLA) [3,58], Polycaprolactone (PCL) [59] and Polyethylene Terephthalate (PET) [60] which presented for *a_w_* = 0.9 at *T* = 25 °C a value of mass gain at equilibrium of 1%, 0.5% and 0.9% respectively. These obtained values of water mass gain for PBS highlighted a hydrophobic character regarding other polymers known as hydrophilic like plasticized starch [61], chitosan [29], or polyamide 6 (PA6) [62], which presented for *a_w_* = 0.9 at *T* = 25 °C a value of mass gain at equilibrium of 50%, 45% and 12 % respectively.

In presence of GnP, whatever the amount, the mass gain at equilibrium were quite similar until *a_w_* ≤ 0.7, and slightly differed for *a_w_* > 0.7. Considering the crystalline part of PBS matrix and the amount of GnP which are both considered as impermeable to water molecules, the mass gain at equilibrium were calculated as a function of the amorphous phase of the polymer matrix using the crystalline index determined from DSC analysis. A single curve was obtained, so it can be concluded that the presence of GnP had no impact on the water sorption mechanism and the water sorption phenomenon occurred in the amorphous part of the PBS matrix. This further confirms a good interfacial adhesion between nanofillers and the polymer matrix.

To go further in the sorption mechanism understanding, the average number of water molecules sorbed in a single amorphous unit of polymer (*Ni*), was calculated from the following equation:(11)$$N_{i}=M_{a}\frac{M_{p}}{M_{w}}$$
where *M_a_* is the mass gain at equilibrium of PBS amorphous part, *M_P_* and *M_w_* are respectively the molar mass of the studied polymer unit (*M_p_* = 172 g mol^−1^) and the molar mass of water (*M_w_* = 18 g mol^−1^).

The evolution of *Ni* as a function of the water activity is presented in Figure 7B. The obtain isotherm logically displayed the same shape than the isotherm presented in Figure 7A. From this representation, it can be seen than for *a_w_* = 0.9, there was one water molecule in average sorbed every 4 units of PBS in the amorphous phase.

To get a better understanding of the water sorption at the molecular scale, the isotherm curve of the neat PBS was modeled using the GAB equation, combined with the theory from Zimm and Lundberg.

The values of GAB parameters and mean relative percentage of deviation modulus (*MRD*) are given in Table 6. Firstly, examination of *MRD* indicated that the model is convenient and allows an accurate description of the experimental sorption isotherm as shown by the theoretical curve plotted in Figure 7B.

From *M_m_* value, it could be deduced that for one amorphous PBS unit, 0.045 H_2_O molecules were in strong interactions, corresponding to an average of one water molecule every 22 amorphous units of PBS. It should be noted that the saturation of the polymer monolayer occurred for *a_w_* ≈ 0.4.

By using the theory of Zimm and Lundberg, it was possible to determine the *MCS* values from the parameters deduced from GAB equation (Equation (10)). The plot of *MCS* versus the water activity is represented in Figure 8.

In Figure 8A, *MCS* values are close to unity at low water activity (below *a_w_* = 0.4) and then increased at higher activities. Beyond *a_w_* = 0.4, interactions for a water molecule to another sorbed molecule appeared and became preponderant leading to the progressive formation of water clusters. At the highest activity (*a_w_* = 0.9), there was about 5 water molecules per cluster. This value was higher than that found for on PA6 [63], chitosan [29] and starch [61] which were 3, 4 and 2 respectively. This higher size of *MCS* for PBS compare to these others hydrophilic polymers can be explained by a lower affinity of water molecules to the PBS chain. As a consequence, at higher water activity, water molecules preferred to be in self-interactions that interact to the polymer chains.

The evolution of *MCS* as a function of *Ni* is presented in Figure 8B. The value of MCS was constant and close to unity until a value of *Ni* = 0.045 then linearly increased. The first part (*Ni* < 0.045) corresponded to an individual distribution of the water molecules and the saturation of the monolayer and the second part (*Ni* > 0.045) corresponded to the formation of water clusters. The water clusters size was proportional to the amount of sorbed water molecules.

The number of sorption sites per monomer unit in the amorphous phase which is defined as the ratio between the number of molecules sorbed per monomer unit in the amorphous phase to the mean cluster size (*N_i_*/*MCS*) was also determined. The evolution of *Ni/MCS* as a function of the water activity is represented in Figure 8C. At low water activity, a linear increase of *Ni/MCS* was observed. As explained previously, in that range of water activity an individual distribution of the water molecules occurred on the different monomer unit of PBS in the amorphous phase. This sorption mechanism was a Henry type sorption with a random absorption of the water molecules in the polymer. For higher water activity, *Ni/MCS* reached a plateau due to the aggregate phenomenon. It can be concluded that the average number of sorption’s sites was statistically one every 22 monomer units of PBS in the amorphous phase. Sabard et al. [64] and Blanchard et al. [57] obtained one sorption site every 22 and 13 monomer units for water sorption for PA6 and EVOH respectively. These differences can be explained by a difference of affinity between water molecules and the respective polymer.

For all systems, *n* values close to 0.5 were obtained for almost all activities (Appendix A in Appendix A). The diffusion mechanism can be considered as Fickian. The water diffusion coefficient values (*D*) were plotted as a function of water activity (*a_w_*) in a semi-logarithmic scale, Figure 9A. The obtained *D* values of the neat PBS were in agreement with those reported in literature [3].

Whatever the film composition, the diffusion rate was dependent on the amount of water molecules sorbed. Constant *D* value was recorded up to *a_w_* of 0.7, after that *D* values decreased. This evolution of *D* was in accordance with the sorption isotherm shape of curve. The constant value of *D* was explained by the Henry’s sorption mode and the was attributed to the water clustering phenomenon [65].

Generally, decrease of *D* is observed when first water molecule clusters are formed. In our case, decrease of *D* should have been observed from *a_w_* = 0.5. This delay could be explained by a low amount of sorbed water molecules in the polymer associated to a low number of sorption’s sites per monomer unit in the amorphous phase. The decrease of *D* was obtained when the *MCS* was higher than 2 as shown in Figure 9B.

The influence of introduced GnP nanofillers on water sorption kinetics was investigated and this effect could be evidenced in Figure 9A. Whatever the nanofiller amount, in the whole range of water activity, the shape of the curve relative to the nanocomposites was similar compared to that of the neat matrix. A decrease of diffusivity was observed in the whole range of water activity. This decrease seemed to be slightly increased as the amount of GnP increased. As graphene act as impermeable obstacles, the diffusion rate became slower because the water molecules follow a more tortuous path to pass through the composite film.

The tortuosity factor can be expressed by the following equation:(12)$$\frac{D}{D_{0}}=\frac{1}{\tau }$$
where *D*_0_ is the diffusion coefficient in the neat polymer, *D* the diffusion coefficient in the composite and *τ* is the tortuosity factor. The tortuosity factor is also defined by the ratio τ=d′d where *d^′^* the length pathway of the permeant molecule in the polymer with impermeable barrier (filler, crystallite, etc) and *d* the length pathway of the permeant molecule in the polymer without impermeable barrier. In this case, since the crystallinity index was not modified by addition of filler, the modification of the tortuosity was considered to be only due to the presence of GnP.

The tortuosity factor *τ*, was determined on the whole range of water activity. For a given system, *τ* was independent of the water activity. Taking uncertainties into account, the *τ* increased as the amount of GnP increased from 1.3 ± 0.2 to 2.0 ± 0.1 for 0.1 wt.% and 2 wt.%, respectively. This indicates that the diffusion mechanism resulted on a geometric type phenomenon.

### 4.4. Water and Dioxygen Permeability

Water permeability measurements were performed on neat matrix and different nanocomposites for water activity range from 0.5 to 1 at 25 °C. The evolution of the water permeability coefficient$\;(P_{H_{2}O})$ as a function of the water activity is shown in Figure 10A. The obtained value of $P_{H_{2}O}$ for neat PBS matrix at *a_w_* = 1 was equal to 2518 Barrer, in good agreement with this reported in the same condition by Charlon et al. (2616 Barrer) [3]. The obtained value was higher than those of common polyesters such as PLA, PET and PHBV 1957 Barrer [58], 150 Barrer [66] and 149 Barrer [67], respectively. In the tested range of water activity, $P_{H_{2}O}$ increased linearly as the water activity increased. This result can be explained by the small plasticization effect of the PBS matrix due to the presence of water molecules sorbed by the polymer. Even with the low hydrophilic character of PBS, the amount of water molecules sorbed increased as seen previously from sorption analyses with the increase of water activity. Then, the presence of water molecules led to increase the mobility of the polymer chains resulting in an increase of water permeability coefficients.

In order to study more specifically the effect of the presence of GnP on the water permeability, the data of Figure 10A are examined. The introduction of GnP led to an improvement of the barrier properties. The reduction of the permeability increased as the GnP amount increased within the PBS matrix. As reported in the literature, GnP are considered as impermeable fillers for small molecules [68,69]. Furthermore, as shown by the TEM micrographs (Figure 2), homogenously dispersed small nanoplatelets within the matrix were observed leading to a significant increase of the gas pathway by a tortuosity effect. As generally observed, the tortuosity increased as the GnP amount increased in the matrix [8,12]. To discuss more specially the effect of the presence of GnP, the values of water relative permeability ($P_{rH_{2}O}$) which is defined as the ratio of the permeability coefficient of the nanocomposite on the permeability of its associated matrix were determined. The evolutions of $P_{rH_{2}O}$ as a function of the water activity are shown in Figure 10B.

In agreement with the previous discussion, $P_{rH_{2}O}$ decreased as the GnP amount increased in the whole range of water activity. Considering the uncertainty of measurements, $P_{rH_{2}O}$ seemed to be constant in the whole range of water activity. From this, it could be concluded that the increase of water sorbed molecules in the polymer did not lead to a plasticization effect of the polymer matrix great enough to minimize the contribution of interfaces between GnP fillers and the polymer matrix at high water activity.

Dioxygen permeability measurements were performed on neat matrix and different nanocomposites for water activity range from 0 to 0.9 at 25 °C. The evolution of the dioxygen permeability coefficient ($P_{O_{2}}$) as a function of the water activity is shown in Figure 11A. $P_{O_{2}}$ of neat PBS in the anhydrous state was equal to 0.135 Barrer. This value was in the same order with the one reported by Messin et al. who obtained a dioxygen permeability of 0.196 Barrer in the same experimental condition [42]. This value was higher than common polyesters such as PET and PHBV which have water permeability of respectively, 0.09 Barrer [66] and 0.031 Barrer [67] but was slightly smaller than the value found on PLA of 0.23 Barrer [58]. As in case of water permeability analysis, in the tested range of water activity, $P_{O_{2}}$ increased linearly as the water activity increased and can be explained by a plasticization of the PBS matrix due to the presence of water molecules sorbed by the polymer which tended to decrease the cohesive density energy of the films. Similar behaviors have been reported in the literature by Tenn et al. on PLA [58].

Here again, introduction of GnP led to an improvement of the dioxygen barrier properties. Improvement increased as the amount of GnP within the PBS matric increased. The evolution of the values of dioxygen relative permeability ($P_{rO_{2}}$) as a function of the water activity are shown in Figure 11B. Considering the uncertainty of measurements, $P_{rO_{2}}$ seemed to be constant in the whole range of water activity. As explained previously, the increase of water sorbed molecules in the polymer did not lead to a plasticization effect of the polymer matrix great enough to minimize the contribution of interfaces between GnP fillers and the polymer matrix at high water activity.

Different models, such as Nielsen [70], Cussler-Aris [71], Bharadwaj [72], have been proposed in the literature to describe the tortuosity and as a consequence the improvement of barrier properties induced by the dispersion of impermeable nanoplatelets fillers in a polymer matrix. The Bharadwaj model gives a good understanding on the modeling of permeability on nanocomposites (Equation (13)) [72].
(13)$$P_{r}=\frac{1-\varphi }{1+\frac{\alpha \varphi }{2}\cdot \frac{2}{3}\cdot \left(S+\frac{1}{2}\right)}$$
where *P_r_* is the relative permeability, *α* the aspect ratio, *φ* is the volume fraction of impermeable phase and *S* is the orientation of fillers in the nanocomposites. A value of *S* = −0.5 showing a perpendicular orientation of fillers with the membrane surface, a value of *S* = 0 a random orientation of fillers and a value of *S* = 1 a parallel orientation of fillers with the membrane surface.

From Bharadwaj model, the mean aspect ratio was calculated and an example of obtained results is shown in Figure 12. For both water and oxygen molecules, an average aspect ratio of 280 ± 50 was found considering the obtained permeability values for all water activity studied and the best fitting was obtained with a value of *S* equal to 1, enhancing the parallel orientation of the film observed on Figure 2. With a value of *S* = 1, the Bharadwaj model becomes the Nielsen model [70]:(14)$$P_{r}=\frac{1-\varphi }{1+\alpha \frac{\varphi }{2}}$$

This model gave the best but was not greatly fitting with our experimental data due an apparent threshold value of 1 wt.% GnP. After this value, the improvement was no longer efficient because of the aggregation of GnP. Graphite base filler are known to have low critical filler concentration of agglomeration, commonly under 1 wt.% [73]. Similar phenomenon has been reported with graphite-based filler, with sometime an increase after the threshold value due agglomeration of the filler which create a connecting pathway of free volume at the interface filler/matrix which ease permeant molecules diffusion [74].

## 5. Conclusions

Nanocomposites were prepared via melt process from a biodegradable polymer matrix (PBS) and low amount of organic lamellar nanofillers (GnP). The influence of the presence of GnP on the crystalline microstructure, polymer chains mobility of amorphous phase, thermal properties, mechanical properties and transport properties was investigated. A rather good GnP dispersion state was evidenced by TEM analyses with the coexistence of small dispersed graphene layers and tactoids. DSC and DMA analyses have shown that the presence of GnP whatever the amount did not change the chain mobility of the amorphous phase of the polymer matrix. Form DSC and WAXS analyses, it has been showed that the crystallinity index *Χ_c_* remained similar after the addition of GnP. Thermal stabilities of nanocomposites were also close to the neat PBS and a slight reinforcement was observed on PBS after addition of GnP. Water sorption analysis was performed. The presence of GnP did not change the water capacity of neat PBS. A detailed analysis of the sorption phenomenon was performed from Guggenheim, Andersen and De Boer (GAB) model combined with Zimm-Lundberg theory. The number of sorption site per unit of PBS and the mean cluster size over the whole range of water activity was determined. In a kinetic point of view, a decrease of diffusion coefficient was observed and was attributed to the presence of good interface between the GnP and PBS matrix and especially to an increase of tortuosity in presence of GnP. This result was confirmed by both water and dioxygen permeability analysis. An improvement of the water and dioxygen permeability of 38% and 35% respectively by addition of 2 wt.% of GnP were obtained respectively. This improvement was attributed to a purely geometric effect with increasing the tortuosity. A study of the impact of the humidity on both water and dioxygen permeability concluded that the plasticization effect of the polymer matrix was not enough to minimize the contribution of the interface between the GnP fillers and the polymer matrix at high water activity. GnP aspect ratio was determined from Nielsen model and showed an average value of 280 ± 50.

## Figures and Tables

**Figure 1 membranes-11-00151-f001:**
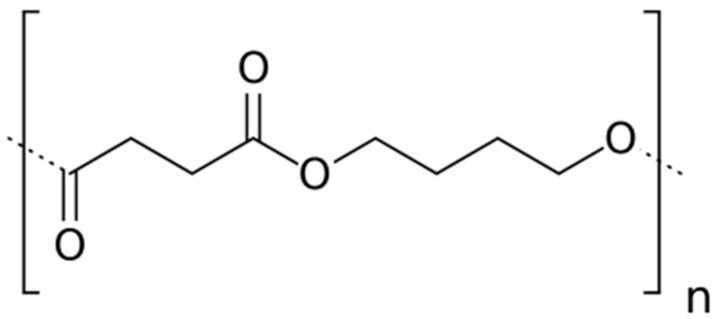
Chemical structure of Polybutylene Succinate (PBS) repeating unit.

**Figure 2 membranes-11-00151-f002:**
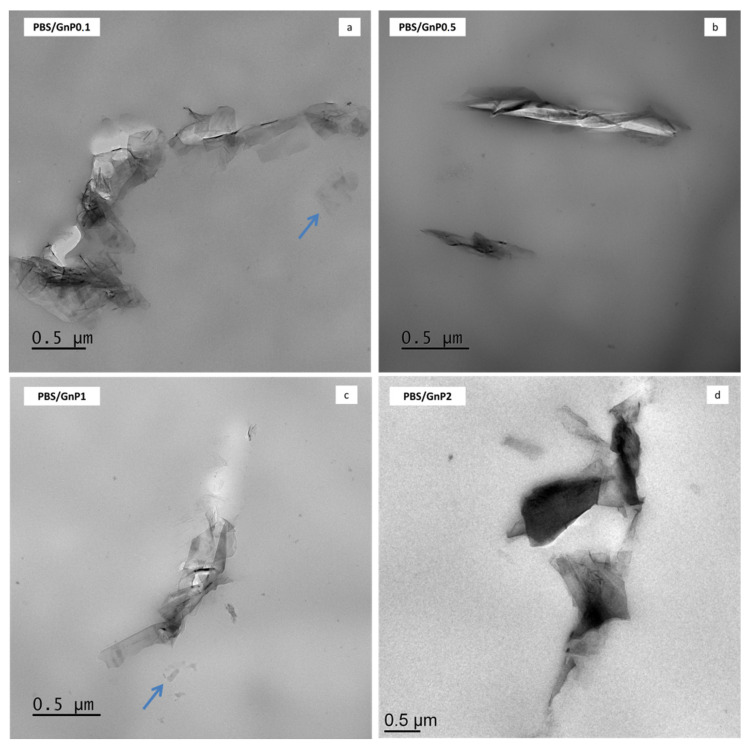
Transmission electron micrographs of Polybutylene Succinate/Graphene nanoplatelets (PBS/GnP) nanocomposites films (**a**) PBS/GnP0.1; (**b**) PBS/GnP0.5; (**c**) PBS/GnP1; (**d**) PBS/GnP2.

**Figure 3 membranes-11-00151-f003:**
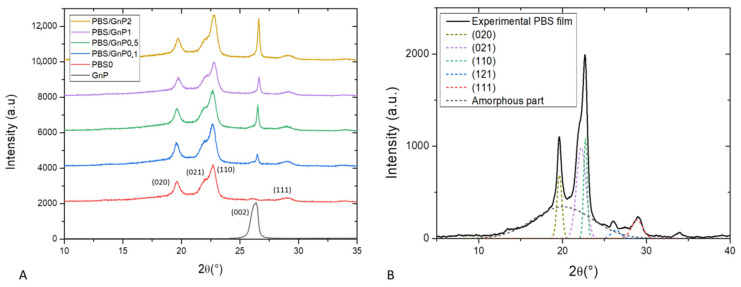
(**A**): WAXS patterns of GnP powder, PBS neat matrix and the different nanocomposites films and (**B**): deconvolution of WAXS patterns and assignation of the different peaks for neat PBS matrix.

**Figure 4 membranes-11-00151-f004:**
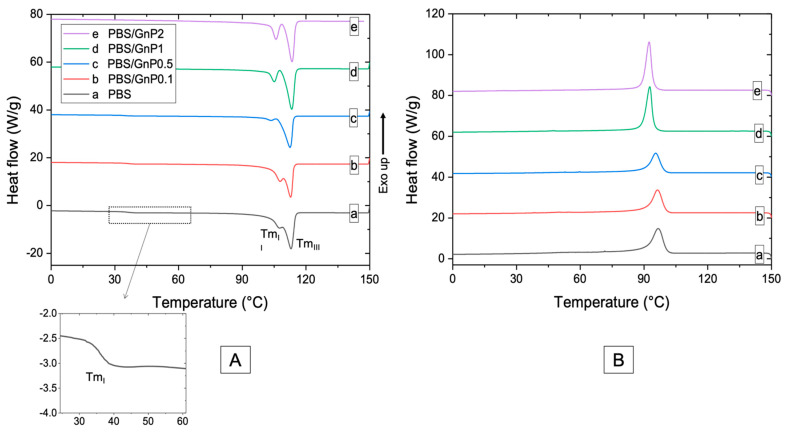
DSC thermograms of (**A**) first heating scan (**B**) cooling scan of neat PBS and corresponding composites.

**Figure 5 membranes-11-00151-f005:**
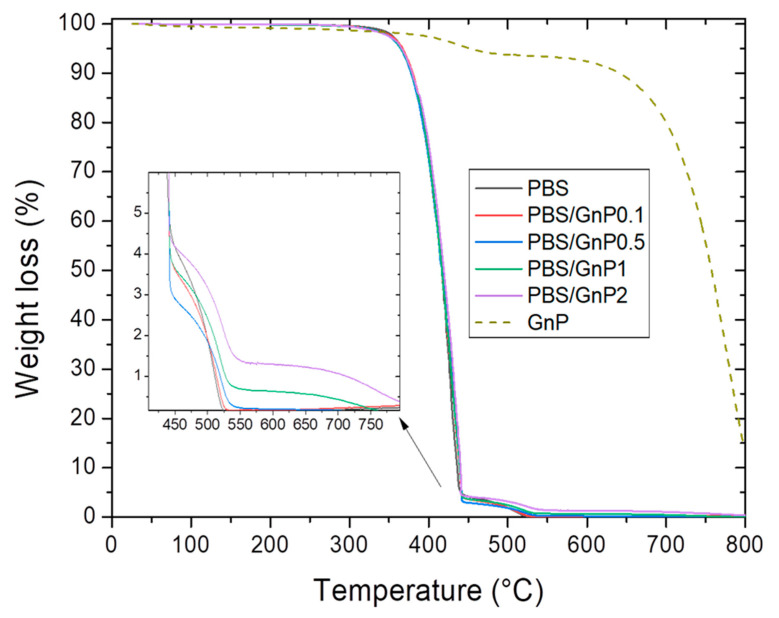
Evolution of the weight loss as a function the temperature for PBS and corresponding composites.

**Figure 6 membranes-11-00151-f006:**
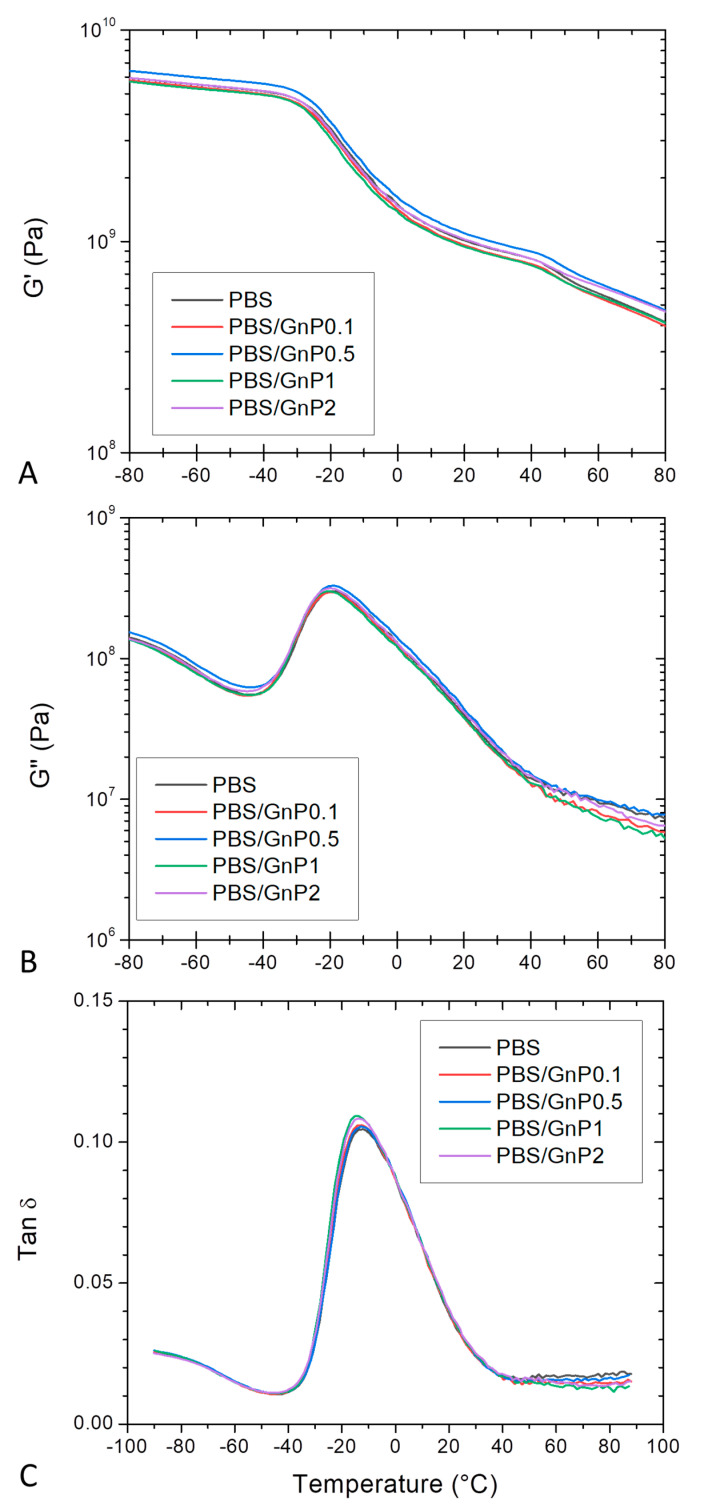
Evolution of (**A**) storage modulus (G’), (**B**) loss modulus (G^″^) and (**C**) tan δ as a function of the temperature of PBS and corresponding composites, at 10 Hz.

**Figure 7 membranes-11-00151-f007:**
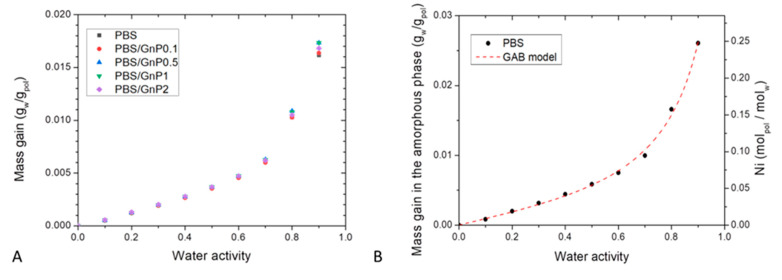
(**A**): Sorption Isotherm curves at T = 25 °C of the neat PBS matrix and the different nanocomposites and (**B**): mass gain in the amorphous phase of neat PBS with GAB model as function of the water activity and average number of water molecules sorbed per amorphous unit of PBS (Ni) as function of water activity.

**Figure 8 membranes-11-00151-f008:**
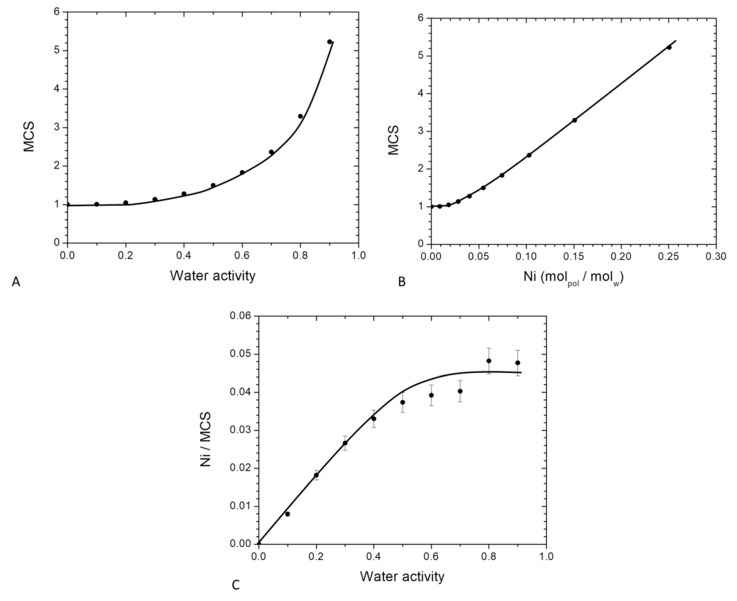
Evolution for neat PBS of Mean cluster size as a function of: (**A**) water activity and (**B**) average number of water molecules sorbed in a single amorphous unit of polymer and (**C**): evolution of Ni/MCS as a function of the water activity for neat PBS.

**Figure 9 membranes-11-00151-f009:**
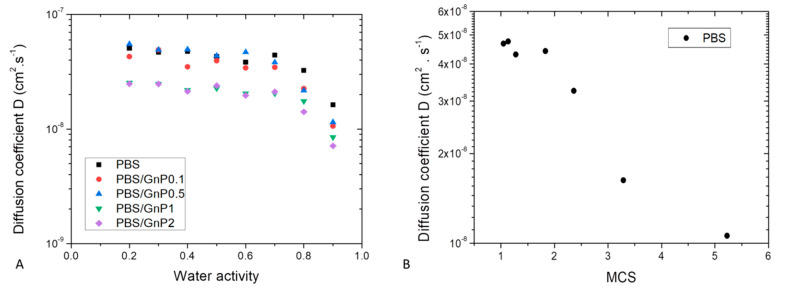
(**A**) Evolution of diffusion coefficient D as a function of water activity for neat PBS and associated nanocomposites. (**B**) Evolution diffusion coefficient as a function of Mean Size Cluster for neat PBS.

**Figure 10 membranes-11-00151-f010:**
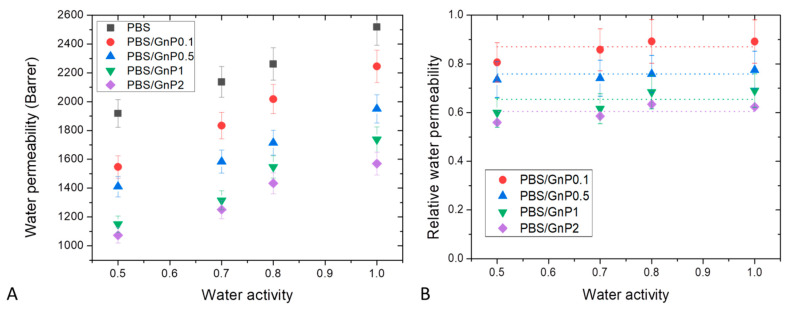
Evolution of the water permeability (**A**) and relative water permeability (**B**) as a function of water activity of neat PBS and corresponding composites.

**Figure 11 membranes-11-00151-f011:**
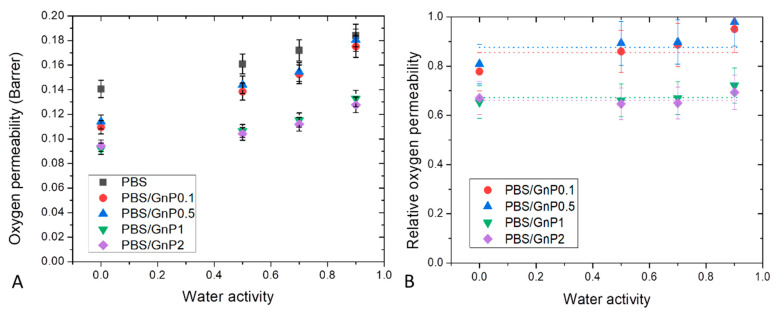
Evolution of the dioxygen permeability (**A**) and relative dioxygen permeability (**B**) as a function of water activity of neat PBS and corresponding composites.

**Figure 12 membranes-11-00151-f012:**
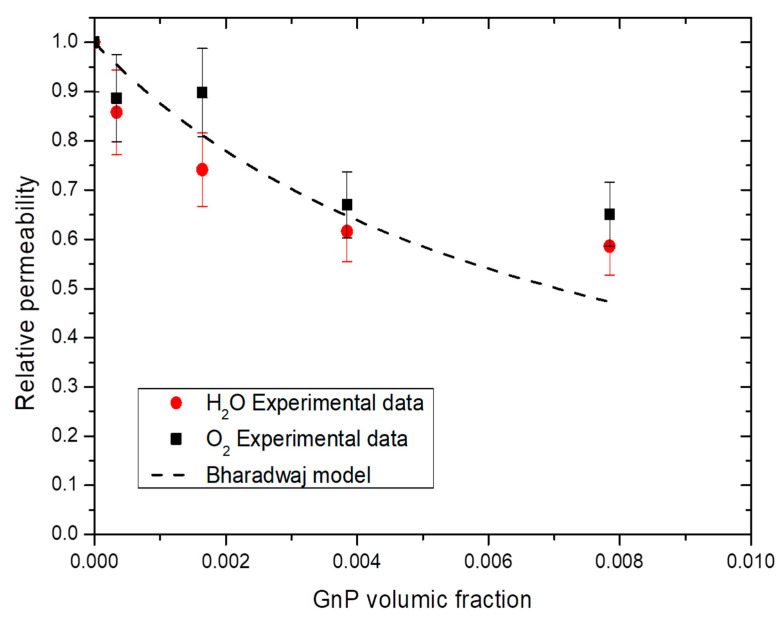
Evolution of the water and oxygen relative permeability as a function of GnP volume fraction for a water activity = 0.7 and Bharadwaj model.

**Table 1 membranes-11-00151-t001:** Theoretical amount and sample code of the different Polybutylene Succinate/Graphene nanoplatelets (PBS/GnP) films.

Theoretical GnP Amount (wt.%.)	Nomenclature
0	PBS
0.1	PBS/GnP0.1
0.5	PBS/GnP0.5
1	PBS/GnP1
2	PBS/GnP2

**Table 2 membranes-11-00151-t002:** Values of number average molecular weight ($\bar{M_{n}}$ ), and dispersity index (*Đ*) of PBS and corresponding composites.

	$\bar{M_{n}}$ $\left(\times 10^{-4}\right)$	*Đ*
PBS	6.79 ± 0.23	2.15 ± 0.08
PBS/GnP0.1	6.60 ± 0.22	2.19 ± 0.08
PBS/GnP0.5	7.03 ± 0.21	2.18 ± 0.07
PBS/GnP1	5.77 ± 0.25	2.09 ± 0.09
PBS/GnP2	5.42 ± 0.32	1.83 ± 0.12

**Table 3 membranes-11-00151-t003:** Values of *Tg*, *Tc*, *Tm* (I, II, III), full width at the half height maximum of the crystallization peak (*FWHM) T_onset_*, Δ*T* and *X_c_* of neat PBS and corresponding composites (Determined by DSC).

		PBS	PBS/GnP0.1	PBS/GnP0.5	PBS/GnP1	PBS/GnP2
First heating	*T_g_* (°C)	−35 ± 1	−36 ± 1	−36 ± 2	−36 ± 2	−38 ± 1
	*Tm _I_* (°C)	35 ± 1	36 ± 1	35 ± 1	34 ± 1	33 ± 1
	*Tm _II_* (°C)	106 ± 1	107 ± 1	103 ± 1	106 ± 1	105 ± 1
	*Tm _III_* (°C)	113 ± 1	113 ± 1	112 ± 1	113 ± 1	113 ± 1
	*X_C_-DSC* (%)	38 ± 1	39 ± 2	37 ± 1	39 ± 1	39 ± 1
Cooling	*T_c_* (°C)	97 ± 1	97 ± 1	96 ± 1	93 ± 1	92 ± 1
	*T_onset_* (°C)	104 ± 1	103 ± 1	103 ± 1	100 ± 1	99 ± 1
	*FWHM* (°C)	5 ± 1	5 ± 1	5 ± 1	3 ± 1	3 ± 1
	Δ*T*	7 ± 1	7 ± 1	8 ± 1	7 ± 1	7 ± 1

**Table 4 membranes-11-00151-t004:** Theoretical, determined GnP loading and thermal degradation temperatures of neat PBS and corresponding PBS/GnP nanocomposites.

Sample	Theoretical GnP Loading (wt.%)	Determined GnP Loading (wt.%)		*T_d5%_* (°C)	*T_d50%_* (°C)	*T_d90%_* (°C)
		Weight %	Volume %			
PBS	0	0.00 ± 0.00	0.00 ± 0.00	367 ± 1	415 ± 2	435 ± 2
PBS/GnP0.1	0.1	0.06 ± 0.02	0.03 ± 0.01	368 ± 1	417 ± 2	439 ± 3
PBS/GnP0.5	0.5	0.28 ± 0.12	0.16 ± 0.06	365 ± 2	417 ± 3	439 ± 3
PBS/GnP1	1	0.66 ± 0.05	0.38 ± 0.03	366 ± 1	415 ± 2	438 ± 2
PBS/GnP2	2	1.35 ± 0.02	0.78 ± 0.01	366 ± 1	419 ± 3	439 ± 3

**Table 5 membranes-11-00151-t005:** Values of Young Modulus, strain and elongation at break for PBS and corresponding composites.

	Young Modulus (MPa)	Stress at Break (MPa)	Strain at Break (mm/mm)
PBS	230 ± 40	45 ± 7	0.35 ± 0.06
PBS/GnP0.1	260 ± 30	50 ± 4	0.34 ± 0.03
PBS/GnP0.5	210 ± 20	40 ± 2	0.29 ± 0.03
PBS/GnP1	260 ± 30	46 ± 3	0.27 ± 0.02
PBS/GnP2	260 ± 30	38 ± 6	0.23 ± 0.04

**Table 6 membranes-11-00151-t006:** Value of the parameters deduced from GAB model considering the sorption isotherm of the amorphous part of the neat PBS.

	*M_m_*	*C_G_*	*K*	*MRD* (%)
PBS	4.70 ± 0.01 × 10^−3^	2.1 ± 0.1	0.93 ± 0.02	5.2

## Data Availability

The data presented in this study are available as Supplementary Material.

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
