# Peer review of "Improvement of Barrier Properties of Biodegradable Polybutylene Succinate/Graphene Nanoplatelets Nanocomposites Prepared by Melt Process"

_membranes, 2021, doi:10.3390/membranes11020151_

Round 1

Reviewer 1 Report

Cosquer et al. reported the manuscript entitled ‘‘Improvement of Barrier Properties of Biodegradable Polybutylene Succinate/Graphene Nanoplatelets Nanocomposites Prepared by Melt Process.’’ The author studied morphology, mechanical properties, water and dioxygen permeability of PBS/GNP composite in detail. This manuscript needs major revision before publication. Some comments are as follow:

  1. Please mention the novelty of the current study.
  2. The introduction section should be a comparative study with some carbon nanofiller-based composites articles. The author should cite the following papers.
  3. ‘‘Graphene nanoplatelet-reinforced poly (vinylidene fluoride)/high-density polyethylene blend-based nanocomposites with enhanced thermal and electrical properties’’.
  4. ‘‘Evident improvements in the rigidity, toughness, and electrical conductivity of PVDF/HDPE blend with selectively localized carbon nanotube’’.
  5. Please explain TEM sample preparation clearly. How much thickness film sample was used for cryo-ultramicrotomy? Is it easy to prepare TEM sample for the film sample?
  6. Please check figure 2b inset, Why plans?
  7. Which ASTM standard was used for mechanical tests (tensile and DMA)
  8. Please change to ??5%, ??50%, and ??90% in Table 4.
  9. Please check Table 6 better to combine with the other Table.

Author Response

First of all, we would like to thank you for the time that you took to review the first version of our manuscript. Following your comments, corrections have been done. The changes made, have been marked in the manuscript following your recommendation.

Reviewer #1

  • Please mention the novelty of the current study.

A: The novelty of this paper deals with the understanding of the water sorption mechanism of PBS polymer in a thermodynamic point of view using phenomenological models (GAB model and Zimm-Lundberg theory). We tried also to correlate the obtained results of structural characterization and the obtained results of transport properties. Moreover, dioxygen and water permeation analyses were also performed as an additional and original route to probe the architecture of the different films in order to evaluate the influence of the presence the graphene fillers on the transport properties of PBS nanocomposites films. It is an original approach to establish the relationships between the nanostructure, morphology, and transport properties of biodegradable based materials. To our best knowledge, there are few publications that deal with the study of transport properties of PBS/Graphene nanoplatelets nanocomposites elaborated by melt process.

This paragraph was added at the end of the introduction part.

  • The introduction section should be a comparative study with some carbon nanofiller-based composites articles. The author should cite the following papers. ‘‘Graphene nanoplatelet-reinforced poly (vinylidene fluoride)/high-density polyethylene blend-based nanocomposites with enhanced thermal and electrical properties’’. ‘‘Evident improvements in the rigidity, toughness, and electrical conductivity of PVDF/HDPE blend with selectively localized carbon nanotube’’.

A: As suggested by the reviewer, the recent studies of Behera et al. have been considered and the following references have been added to the manuscript:

Reference #14: “Behera, K.; Chiu, F.C. Evident improvements in the rigidity, toughness, and electrical conductivity of PVDF/HDPE blend with selectively localized carbon nanotube. Polym. Test. 2020, 90, 106736, doi:10.1016/j.polymertesting.2020.106736.”

Reference #15: “Behera, K.; Yadav, M.; Chiu, F.C.; Rhee, K.Y. Graphene nanoplatelet-reinforced poly(Vinylidene fluoride)/high density polyethylene blend-based nanocomposites with enhanced thermal and electrical properties. Nanomaterials 2019, 9, doi:10.3390/nano9030361.”

  • Please explain TEM sample preparation clearly. How much thickness film sample was used for cryo-ultramicrotomy? Is it easy to prepare TEM sample for the film sample?

A: The TEM method section has been updated considering the sample preparation. To go further in the explanation, the analyses were realized by an external operator. Films of about 100 µm were holding into a vise clamp during the slicing process. Diamond knives were used to cut slices of about 80 nm thickness. The operation was realized at -90°C to ease the slicing process since PBS glass transition temperature is -35 °C.

  • Please check figure 2b inset, Why plans?

A: Diffraction plans were determined using data from the literature. Modification on the Figure 2a and b (now Figure3a and b) inset have been done for clarity.

  • Which ASTM standard was used for mechanical tests (tensile and DMA)

A: No ASTM standard was used for tensile test analysis. The experiments were performed on films with H3 type tensile specimens. Only the film thickness differed but was measured and was considered for stress determination. However, the tensile test was performed on at least 10 specimens for each sample.

DMA results were acquired on a single specimen with precisely measured (about 17.5 mm x 4.0 mm x 2.0 mm) due to experimental setup.

  • Please change to ??5%, ??50%, and ??90% in Table 4.

A: As suggested by the reviewer, modification of the Table 4 has been done.

  • Please check Table 6 better to combine with the other Table.

A: We understand that Table 6 is small but it is difficult to combine it with another table since it is the only one in the transport properties section. We think the GAB modeling parameters should appear clearly in a table rather than in the core of the text.

Reviewer 2 Report

1.       I would recommend addition of the structures of biodegradable ( Polybutylene Succinate (PBS).).
2.       Half of the references is devoted to the general properties of PBS, not the main topic. I have an impression that the information about physical chemistry of micellization are too detailed for this type of paper.
3. Please mention the advantages of the Polybutyl- 2en Succinate/Graphene Nanoplatelets.
4.The stabilization aspect of Polybutyl- 2en Succinate/Graphene  is not well listed in the review
5.Generally, this review is quite valuable, despite minor drawback mentioned above.
I think the manuscript would be worth of publishing in the Journal providing following remarks/suggestions will be accounted for. I hope that they may help improving the manuscript appropriately. Alternatively, the authors should give reasonable comments and answers.

Author Response

First of all, we would like to thank you for the time that you took to review the first version of our manuscript. Following your comments, corrections have been done. The changes made, have been marked in the manuscript following your recommendation.

Reviewer #2

  • I would recommend addition of the structures of biodegradable ( Polybutylene Succinate (PBS).).

A: As suggested by the reviewer, the chemical structure of PBS was added and displayed as the Figure 1.

  • Half of the references is devoted to the general properties of PBS, not the main topic. I have an impression that the information about physical chemistry of micellization are too detailed for this type of paper.

We do not agree. Only 8 references on the 75 are devoted to the general context of PBS. For the others in the introduction part, one for each concerns the transport properties of polyimide, poly (L-lactic acid), poly (vinyl alcohol) nanocomposites, and all the others concern PBS nanocomposites.

  1. Please mention the advantages of the Polybutyl- 2en Succinate/Graphene Nanoplatelets.

A: As indicated in the introduction part, for our concern, the addition of nanofillers with a high aspect ratio within a polymer matrix can lead to an increase of the barrier properties if the fillers are well dispersed and also if the quality of the interfaces between the fillers and the matrix was convenient. As graphene act as impermeable obstacles, the diffusion of the permeant molecules rate became slower because they follow a more tortuous path to pass through the composite film. Besides, due to a higher aspect ratio compared to montmorillonite, the amount of graphene added is smaller.

  • The stabilization aspect of Polybutyl- 2en Succinate/Graphene is not well listed in the review

A: We did not check the effect of the presence of graphene on PBS matrix over a longer time. However, we checked the influence of the presence of Graphene during the melt process. Obtained results are discussed in the manuscript. Moreover, all experimental measurements were performed in the same period time to avoid a possible effect of degradation of PBS matrix in presence of the fillers.

  • Generally, this review is quite valuable, despite minor drawback mentioned above.

A: OK

Reviewer 3 Report

This study intitled "Improvement of barrier properties of biodegradable Polybutylene succinate/Graphene nanoplatelets nanocomposites prepared by melt process" is interesting for biodegradable nanocomposite based packaging film and the authors have presented and interpreted their finding in a high level. 

I suggest the publication after a minor revision of introduction sections as followsQ

Please seperate the paragraph reffered to carbon based nanofiilers form platelet based nanoffiller. Give more information for the prospect of using carbon based nanoffilers and especially graphene oxide nanofiilers in food packging.

Author Response

First of all, we would like to thank you for the time that you took to review the first version of our manuscript. Following your comments, corrections have been done. The changes made, have been marked in the manuscript following your recommendation.

Reviewer #3

  • Please seperate the paragraph reffered to carbon based nanofiilers form platelet based nanoffiller.

A: As advice by the reviewer separation was done

  • Give more information for the prospect of using carbon based nanoffilers and especially graphene oxide nanofiilers in food packging.

A: The prospect of food packaging can be considered as the end of this study but is not discussed in this paper. Since there is, at this time, no study on the migration of GnP in the food for an application of food packaging for these types of membranes (PBS/GnP) has been referenced. However, some paper discussed the possibility of nanocomposites such as Manikandan et al. [1] who studied both the barrier improvement of Polyhydroxybutyrate (PHB)/graphene and also conclude insignificant cytotoxicity of such food packaging on food. Goh et al. [2] studied a sandwich-architectured Poly(lactic acid)/Graphene composite in which the graphene layer was protected by being encapsulated between layers of PLA.

Since no cytotoxicity study or migration study was carried in this paper, the authors decided to not mention any end application for this type of membranes. However, a further study carried on the base of this paper should conclude to a potential use of this type of nanocomposite, in a bulk utilization or in a multilayer use.

References:

  1. Manikandan, N.A.; Pakshirajan, K.; Pugazhenthi, G. Preparation and characterization of environmentally safe and highly biodegradable microbial polyhydroxybutyrate (PHB) based graphene nanocomposites for potential food packaging applications. Int. J. Biol. Macromol. 2020, 154, 866–877, doi:10.1016/j.ijbiomac.2020.03.084.
  2. Goh, K.; Heising, J.K.; Yuan, Y.; Karahan, H.E.; Wei, L.; Zhai, S.; Koh, J.-X.; Htin, N.M.; Zhang, F.; Wang, R.; et al. Sandwich-Architectured Poly(lactic acid)–Graphene Composite Food Packaging Films. ACS Appl. Mater. Interfaces 2016, 8, 9994–10004, doi:10.1021/acsami.6b02498.

Round 2

Reviewer 1 Report

The authors have addressed all the major concerns, so I recommend it for publication.